# Factors Influencing Consumer Purchase Intentions of Organically Grown Products in Shelly Centre, Port Shepstone, South Africa

**DOI:** 10.3390/ijerph16060956

**Published:** 2019-03-17

**Authors:** Siphelele Vincent Wekeza, Melusi Sibanda

**Affiliations:** Department of Agriculture, Faculty of Science & Agriculture, University of Zululand, KwaDlangezwa 3886, South Africa; sphehvwekeza@gmail.com

**Keywords:** multiple regression model, organically grown products, perceptions, purchase intentions, Shelly Centre

## Abstract

In the last few years, the market of organically grown products (OGPs) has continued to grow due to speculated concerns for the environment, food safety and health issues. The market for OGPs in South Africa appears to be under threat; with their demand outstripping their supply. In light of this background, there are relatively few studies on the consumer purchase intentions of OGPs in South Africa, and thus, less understanding about its demand market drivers. The purpose of this paper is to identify the factors influencing the purchase intentions of OGPs (fruits and vegetables) in Shelly Centre in Port Shepstone in Kwa-Zulu Natal Province of South Africa. Using a quantitative descriptive cross-sectional research design, a hundred and fifty (150) OGP consumers were selected through a systematic random sampling technique from three accredited OGP retail outlets namely Pick n’ Pay, Spar and Woolworths. Generally, descriptive results show that the interviewed consumers in Shelly Centre were reasonably educated and knowledgeable about OGPs. A higher proportion of the interviewed consumers in Shelly Centre consisted of women, employed and not of African descendant (ethnic group) consumers. Most were confident that OGPs are environmentally friendly, safe, high-quality products, and have a better taste compared to conventionally grown food products. A somewhat fair proportion expressed mixed feelings concerning the belief that OGPs are priced higher, their appeal to nature (smell), and their willingness to purchase OGPs regardless of price. Nonetheless, most were adamant that OGPs are difficult to find on the market. A multiple regression model analysis results reveal that consumer demographics; ethnicity (not of African descent) (*p* < 0.001), and monthly household income (*p* < 0.05) are statistically significant and positively influence the consumer purchase intentions of OGPs in Shelly Centre. Conversely, consumer perceptions that OGPs are priced higher (*p* < 0.05), have a better taste and of quality (*p* < 0.001), and the difficulty to find on the market (*p* < 0.001) are statistically significant and negatively influence the consumer purchase intentions of OGPs in Shelly Centre. The findings of this paper stress the need to design strategies and elements (marketing mix) to make OGPs affordable and readily available to consumers. Likewise, consumers from all ethnic groups and income levels need to be conscious of the environmental and health benefits of OGPs to make informed purchase decisions. To promote the purchase of OGPs; from a policy perspective, the government can offer support such as a consumer price subsidy to make OGPs affordable, and the provision of effective regulations and certification around the marketing of OGPs.

## 1. Introduction

The industry of organically grown products (OGPs) in South Africa is growing, which is a newfound aspect of the recent shifts in the food demand in the whole country [1,2,3]. The rapidly rising OGP market has set off many researchers to compare several features of organically and conventionally grown products [1,2]. Due to the spontaneous concerns over conventional food products, consumer demand and preferences of OGPs imply on the agricultural production techniques [4,5,6,7]. However, the awareness of the harmful effects of chemical residues in conventionally grown products is increasing among consumers. As an alternative to these, consumers shift towards OGPs [2,3,8,9,10,11]. 

In South Africa, the demand of OGPs is increasingly becoming more critical as consumer attitudes and preferences influence the direction of food retailers’ strategies [1,3,12]. This market shift is due to consumers’ interests for safe alternatives, exclusively, OGPs [13,14]. Currently, OGPs are perceived as healthy by most consumers as these are made up of natural elements and mostly preferred compared to conventionally grown products. Hence, there has been a belief that OGPs are safer and healthier than conventional products, and there are also numerous consumers who are willing to pay large price premiums for them [15,16]. 

Currently, there is little knowledge of South African consumers’ concerns and attitudes towards OGPs [12,17,18]. Studies conducted in South Africa still lack a clear understanding of consumer perceptions and their purchase intentions towards OGPs [3,12]. How consumers make choices in purchasing food is different and very complex. Hence, consumers are worried about various issues; thus, the circumstances cannot be definite that consumer behaviour towards OGPs is a straight forward issue [19]. Regarding environmental concerns, the relationship about consumer attitude and behaviour is not surely straightforward mostly where the products point out a conflict between environmental safety and product gains. Many consumers claim they have not derived any benefits from OGPs, which may threaten the OGPs market and the available literature lacks substantial evidence to justify claims that OGPs are better than conventional products [20]. 

Numerous studies worldwide show that consumers tend to purchase OGP for multiple reasons. These reasons involve sensory, and non-sensory features of the products produced organically [3]. However, several studies have revealed that food safety concerning human health concerns is considered the main influencing factor for consumers to purchase OGPs [21]. Some studies show that concerns have also risen around the matters of what remains in food from insect repellents, fertilisers, or whichever other types of synthetic additives [22], this, therefore, affects the safety of the products. Among the commonly cited cause for the increasing demand of OGPs is environmental consciousness by consumers [23,24]. Many consumers are noted to have become more aware of the environmental, social, and economic impacts of their purchase decisions and choices may lead to [25]. Some studies reveal that product attributes such as “taste” is one of the most crucial factors influencing the purchase of OGPs [2,6,19]. Olson [13]; Rana and Paul [26], and Scalco [10], suggest that since OGPs have a strong association with high prices, many consumers perceive OGPs to be of higher quality compared to conventionally grown products. Hence, this belief elevates consumers’ perceptions that OGPs are quality food products and taste better. 

Understanding consumers’ demand of OGPs in South Africa is increasingly becoming important since consumer attitudes and preferences intensely influence the direction of food retailers’ strategies [1,3], and the promotion of healthy living and environmental protection. In light of this, this paper seeks to understand the factors influencing consumers’ purchase intentions of OGPs in Shelly Centre, which is an upmarket area in Port Shepstone in Kwa-Zulu Natal Province.

## 2. Materials and Methods

### 2.1. Description of the Study Site

The result of the paper is an outcome on research conducted in Shelly Centre in Port Shepstone. The location of Port Shepstone is on the lower south coast of KwaZulu-Natal Province under Ray Nkonyeni Local Municipality, which falls under Ugu District Municipality, in South Africa [27]. The study area was selected using a multistage sampling technique, where out of the nine provinces in South Africa, the selection of Kwa-Zulu Natal Province is random. Out of the ten districts of the Kwa-Zulu Natal Province, Ugu District Municipality is then randomly selected, and the town Port Shepstone and Shelly Centre chosen purposefully. The motivation for the selection of Port Shepstone is because it is the central business area, and Shelly Centre is an upmarket type place in the Ugu District Municipality in the KwaZulu-Natal Province. Consumers of OGPs are usually located within and around the upmarket areas. Figure 1 is a map showing the location of Shelly Centre in Port Shepstone under the Ray Nkonyeni local Municipality.

### 2.2. Research Design

This paper adopts a quantitative research approach. A quantitative research approach provides flexibility in data handling, as well as concerning statistical analysis, comparative studies, and reiterating of data collection, and thus, confirms the stability of instruments used [28]. A cross-sectional research design was employed since it generally uses a survey method to collect data at a single point in time, and hence it is comparatively cheap and consumes less time to conduct but adequately allowing for the capturing of the variables and information of interest.

#### 2.2.1. The Conceptual Framework of the Factors That Influence Consumer Purchase Intentions of Organically Grown Products

Consumers’ purchase intentions or behaviour is generally an attribute of human behaviour. The description of consumers’ behaviour is generally a collection of actions intended to meet the consumption needs of individuals with different personalities. There are many theories of consumer behaviour that include for example; the economic man approach, psychodynamic approach, rising income theory, behaviouristic approach, and cognitive approach. The economic man approach suggests that consumers must be able to distinguish between choices accessible to them when purchasing foodstuffs [29]. The economic man approach asserts that consumers have to rank the available choices according to their level of importance and take the best possible decision [30]. However, this approach does not adequately explain consumer behaviour since some consumers purchase products without giving it much thought. The psychodynamic approach asserts that consumer behaviour is subject to biological influences through intrinsic drives which act on the thought processes of the consumer [31]. The rising income theory outlines that consumer spending habits change with a change in income, implying that as the income increases, spending on most items is more likely to increase [32]. However, the increase in income does not follow the same trend for all consumers. The behaviouristic approach states that behaviour is everything that a person does or displays through being in contact with external events [33,34]. The behaviouristic approach also establishes links to human behaviour such as radical behaviourism (which takes into account feelings, state of mind and introspection) and cognitive behaviourism [35]. The cognitive approach takes into account the actions or traits observed concerning consumer behaviour [36]. The environment in which consumers live in and their social experiences influence the internal decision-making process by the consumer [23]. The cognitive approach, therefore, describes purchasing activities like problem-solving in nature. In the cognitive approach, the consumer solves a problem (arrives at purchasing decision) through collecting information, processing those, and taking the decision guided by that information processing [37]. As a result, a consumer’s knowledge, perception, beliefs, and attitudes conditions what is wanted by a consumer [38]. Therefore, the theory of consumer behaviour, which is an analytical cognitive model guides this paper. In the consumer behaviour theory, the explanatory variables specification is for each person; that is every consumer has a model of consumer behaviour in mind (the factors that shape motivation and behaviour). Based on the consumer behaviour theory, various factors influencing the consumer purchase intentions to purchase OGPs may be due to consumer demographic characteristics (education, age, gender, marital status, ethnicity, household (family) size, monthly household income, employment status) which can shape the knowledge or motivation of consumers to purchase OGPs. Additionally, consumer perceptions towards OGPs for example, perceptions on price, environmental friendliness, food safety, smell, taste and quality, accessibility on the market, and healthiness of the product can influence the attitude of the consumer. Figure 2 shows a conceptual framework of the factors influencing consumer purchase intentions of OGPs.

#### 2.2.2. Study Population, Sample Size and Procedure

The target populations were the fruits and vegetable consumers who shopped in the OGPs retail outlets namely; Pick n’ Pay, Spar and Woolworths in Shelly Centre. According to Du Plooy [39], it is of importance to deal with an adequate sample size to gather precise data about a group. Larger samples are more expressive than smaller samples which decrease the level of accuracy yet moderately accessible and cheap [40]. The sample size in this paper consists of 150 respondents (50 from each retail outlet). Arguably, this is a small sample size that can decrease the level of accuracy and reliability of the data collected. Nonetheless, this sample size is deemed to be adequate to perform statistical analysis and as well as small enough to be manageable. A probability systematic random sampling procedure was employed whereby every 5^th^ consumer was systematically picked from the fruit and vegetable section of the three selected retail outlets as done by Rahman and Noor [41]. The researcher, when collecting data stood by the fruit and vegetable OGPs section (indicated by store boards) and systematically picked the respondents (consumers) who were picking fruits and vegetables from these shelves. Therefore, the assumption here was that the respondents were OGPs consumers.

### 2.3. Data Collection

Before the commencement of data collection, we obtained ethical clearance approval for this research as per the University of Zululand Research Ethics Committee. Additionally, the management of Shelly Centre and that of the three retail outlets (Pick n’ Pay, Spar and Woolworths) approved the research to be conducted within their premises. Data collection was through a pre-tested structured questionnaire (closed and open-ended questions) using a survey method. Pre-testing the questionnaire was done to enhance the validity and reliability of the questionnaire and hence the data. The structured questionnaire collected data from OGP consumers for example; their demographic profile (characteristics), what OGPs meant to them? How they identified or distinguished OGPs on the market from other conventional food products? How frequently they purchased OGPs? Their willingness to purchase OGPs regardless of price premiums attached to OGPs, and their perceptions towards OGPs. The purpose of this data collection was to establish the reasons (factors) behind the purchase of OGPs by consumers in Shelly Centre. Each respondent received a questionnaire, that is every fifth selected consumer who was spotted picking fruits and vegetables in the OGPs section of the three retail outlets in Shelly Centre. The questionnaires were also made available (translated) to the most spoken language in the study area, which isiZulu for ease of those who could not read nor write in English. Data collection prolonged for 5 days after the pay dates; that is the 15th (15th–20th) of the month since many government professionals (nurses and teachers) in South Africa are paid during these times and also on the 25th–30th) which is the end of each month (usually pay dates for other workers). It is during these days that many people would usually shop for groceries after receiving income. Data collection commenced in November/December 2017 and January 2018, during working hours (08 h 00–17 h 00), which usually is the busiest period of each month during and immediately after the payment dates. Face-to-face interviews were conducted to give the respondent clarity on how to answer some questions where necessary. The interviews on average lasted up to 30 min. 

### 2.4. Data Analysis

Following the data collection process, data was cleaned and captured into Microsoft Excel 2016 (Microsoft Corporation, Washington, USA) and exported to SPSS version 25 (SPSS Inc. (IBM), Chicago, Illinois, USA) and STATA version 14 software (StataCorp, Texas, USA) for analysis. Descriptive statistics (frequencies and percentages) is employed to describe the socio-economic characteristics of consumers and their perceptions of OGPs. A cross-tabulation statistic in this paper between socio-demographic characteristics of respondents and the intention to purchase OGPs variable is employed to establish if a statistically significant relationship (association) existed between the variables. This paper employs a Cronbach’s alpha (α) statistic to measure reliability (internal consistency) of a related set of items for the Likert scale data. A multiple regression model analysis is used to identify and assess the factors influencing the purchase intentions of OGPs by consumers in Shelly Centre in Port Shepstone. A correlation Pearson analysis is employed to determine which variables to include in the final multiple regression model analysis. 

#### Specification of the Multiple Regression Model

Multiple regression is a many-to-one modelling and an expansion of the simple linear regression model, where two or more independent variables are used to predict the variance in one dependent variable [42]. 

The specification of the multiple regression model is as follows (Equation (1)):(1)$$y=\beta_{0}+\beta_{1}X_{1}+\beta_{2}X_{2}+........+\beta_{P}X_{P}+\varepsilon$$
where: -

$\beta_{0};+\beta_{1}X_{1}+\beta_{2}X_{2}.....\beta_{P}X_{P}$: are the linear parameters to be estimated.

ε: is the error term.

Then the estimated equation can be specified as follows (Equation (2)):(2)$$Ý=b_{0}+b_{1}X_{1}+b_{2}X_{2}+........+b_{P}X_{P}\;\varepsilon$$
where:

Ý: is the predicted value of the dependent variable.

$b_{0},b_{1},b_{2}\;....b_{p}$: are the estimates of $\beta_{0}+\beta_{1}+\beta_{2\;...........}+\beta_{p}$

In the multiple regression model, while holding all other factors constant, the interpretation of each coefficient is the estimated change in y corresponding to a one-unit change in the dependent variable [43].

The dependent variable in this paper is the consumer purchase intention of OGPs. A purchase intention in this paper is the assumed consumer’s willingness to purchase an OGP. In other words, the purchase intention is a proxy for the actual purchase. The assertion is that a consumer’s intention stimulates or drives the actual purchase of OGPs. What a consumer contemplates buying represents a purchase intention. Although the purchase intention by a consumer is an advance plan that may not necessarily lead to an actual purchase in the future, a positive purchase intention would suggest an increase in the likelihood of purchasing the product and a willingness to pay for the product. The dependent variable (purchase intention) was measured using a 5-point Likert scale (1—very unlikely; 2—unlikely; 3—somewhat likely; 4—likely; 5—very likely). Table 1 presents a summary of the explanatory (independent) variables inputted in the multiple regression model, their description and expected outcome.

The number of schooling years represents the level of education of the consumer in this paper. The study of Yin et al. [44], reveals that the educational levels of consumers marginally influence the purchase intentions of OGPs food. Therefore, the education of the consumer in this paper is hypothesised to have a positive influence on the purchase intention of OGPs. The age group of the consumer is assumed to show different purchase intentions towards OGPs. For example, Mhlophe [3], asserts that a 21-year-old OGPs consumer reacts differently as compared to a 60-year-old consumer. However, different consumers might respond differently concerning their purchase intentions of OGPs even if they are within the same age group. Therefore, this variable is expected to have either a positive or a negative correlation concerning consumer purchase intentions of OGPs. The gender influence of the consumer mostly relies on reasons such as that women are more likely to be anxious about products that they strongly connect to their quality lifestyles. On a different note, the influence of gender may not be predeterminable. In this paper, the gender of the consumer measurement is a dummy variable (that is male is coded with 0 and female with 1). Therefore, the gender of the consumer is hypothesised to have either a negative or positive influence concerning the purchase intentions of OGPs since there are mixed feelings on its influence. Marital status was measured categorically as a dummy variable (being single, divorced or widowed coded with 0 and married with 1). A study of Mhlophe [3], posits that single, divorced or widowed consumers are more likely to purchase OGPs in contrast to married consumers. This assertion could be because of that single, divorced or widowed consumers may be able to afford the price premiums attached to OGPs. On the other hand, married people may not have positive purchase intentions of OGPs since they may have many family members to feed. Therefore, marital status is hypothesised to have either a positive or a negative influence on the purchase intentions of OGPs. Ethnic group analysis is another relevant demographic variable to consider in this paper. The ethnicity of the consumer was measured categorically (that is being of African descent was coded with 0, being of non-African descent (Coloured, White or Asian coded with 1). According to Mhlophe [3], ethnic groups tend to act in a different way when they are forming their purchase intentions. The influence of ethnicity on the purchase intentions of OGPs by consumers may not be conclusive, and thus ethnicity is hypothesised to have either a positive or a negative influence on the purchase intentions of OGPs. Household size in this paper represents the consumer’s family size (household members living with the consumer) at the time of the study, and this was measured categorically. Mhlophe [3], asserts that consumers from a household with a lower family size are more likely to have a positive purchase intention of OGPs. This assertion is because OGPs are expensive, and therefore it would be relatively cheaper to purchase OGPs for a smaller family size compared to a larger one. Therefore, the household size variable is expected to have a negative correlation with the purchase intentions of OGPs. Disposable income is another factor that is considered crucial in influencing purchase intentions of OGPs [3]. Monthly household income was measured categorically. Dettmann and Dimitri [45] posits that the purchase intention of OGPs is likely to increase when consumers’ income increases. Therefore, households with higher income levels are likely to have positive purchase intentions of OGPs since they could afford to pay for price premiums [3]. There is a strong association between the employment status and income levels of consumers. In this paper, employment status is measured as a categorical variable (unemployed = 0, and being employed = 1). Consumers that are part-time or full-time employed are likely to afford OGPs and therefore, are likely to develop positive purchase intentions towards such products [46]. Employment status is therefore hypothesised to have a positive influence on the purchase intentions by consumers of OGPs. Consumer perceptions can also influence their purchase intentions. According to Mhlophe [3], the price of a product is one of the most significant indicators on the market. Therefore, high prices result in a less repeated purchase of OGPs compared to low priced products [23,47]. Therefore, the perception that OGPs are priced higher is therefore hypothesised to have a negative influence on the purchase intentions of OGPs by consumers. Environmental friendless is one of the main motivating factors towards the purchase of many products, inclusive of OGPs [8]. The perception that OGPs are environmentally friendly is therefore hypothesised to have a positive influence on the purchase intentions of OGPs. Food safety is a general fear for a large portion of consumers [9]. The perception that OGPs are safe compared to conventional food products is hypothesised to have a positive influence on the purchase intentions of OGPs. The study of Seljåsen et al. [48], state that OGPs taste better and have a good scent which is more natural. The perception that OGPs have a better smell is therefore hypothesised to have a positive influence on the purchase intentions of OGPs. Studies of Mditshwa et al. [2]; Marian et al. [47] and Bryła [49] reveals that ‘taste’ is one of the most fundamental factors driving the purchase of OGPs. The perception that OGPs have a better taste is therefore hypothesised to have a positive influence on the purchase intentions of OGPs. Organically grown products are believed to be difficult to find (not readily available) on the market. The reason behind the high difficulty to access of OGPs is that there is a shortage of producers [50,51]. The perception that OGPs are difficult to find on the market is therefore hypothesised to have a negative influence on the purchase intentions of OGPs. Health is gradually becoming an important personal and social value aspect [19]. Consumers’ attitude toward OGPs is a direct linkage to health issues [52]. The perception that OGPs are healthier than conventional food products is therefore hypothesised to have a positive influence on the purchase intentions of OGPs. 

## 3. Results

### 3.1. Demographic Characteristics of the Interviewed Consumers in Shelly Centre

The results in Table 2 reveal that all the interviewed consumers in Shelly Centre had at least 12 schooling years (matric education). The mean years of schooling by the interviewed consumers was about 15 years. The age group of consumers is anticipated to reveal different purchase intentions towards OGPs [53]. The results in this paper show that the highest percentage (35%) of the interviewed consumers who purchased fruit and vegetable OGPs were in the age group of between 35 and 49 years (Table 2).

Cross-tabulation also confirms that the most substantial contribution of consumers who agreed to purchase OGPs are from the age group of 35 and 49 years. A Pearson Chi-square statistic (χ^2^ = 0.200) between the age group of the consumer and the intention to purchase OGPs indicates that there is no statistically significant association between these variables. 

The gender influence of the consumers is mostly dependent on reasons such as that women are more likely to be concerned about products that they strongly connect to quality lifestyles. Females constituted a higher proportion (about 63%) of the interviewed consumers (Table 2). Cross-tabulation between the gender of the consumer and the intention to purchase OGPs indicates that the most substantial contribution of consumers who agreed to purchase OGPs are females. A Pearson Chi-square statistic (χ^2^ = 0.602) suggests that there is no statistically significant association between these variables. 

The ethnicity majority (about 59%) of the interviewed consumers in Shelly Centre were not of African descendant (Coloured, White or Asian). Cross-tabulation between the ethnicity of the consumer and the intention to purchase OGPs indicates that the most substantial contribution of consumers who agreed to purchase OGPs belonged to the ethnic group; not of African descendant (Coloured, White or Asian). A Pearson Chi-square statistic (χ^2^ = 0.049) indicates that there is a statistically significant association between these variables. 

The results show that the marital status majority (about 53%) of the interviewed consumers in Shelly Centre were single, divorced or widowed at the time of the study (Table 2). Cross-tabulation between the marital status of the consumer and the intention to purchase OGPs indicates that a somewhat substantial contribution of consumers who agreed to purchase OGPs was single (unmarried). A Pearson Chi-square statistic (χ^2^ = 0.375) suggests that there is no statistically significant association between these variables. 

Employment status has a positive correlation with income levels [54]. The results show that the majority (about 73%) of the interviewed consumers in Shelly Centre were employed (Table 2). Cross-tabulation between the employment status of the consumer and the intention to purchase OGPs indicates that a substantial contribution of consumers who agreed to purchase OGPs was employed. A Pearson Chi-square statistic (χ^2^ = 0.073) suggests that there is no statistically significant association between these variables.

Table 2 shows that a relatively higher proportion (about 45%) of the interviewed consumers had a household size which is about less or equal to 5 members. Cross-tabulation between the household size categories and the intention to purchase OGPs indicates that a considerable contribution of consumers who agreed to purchase OGPs belonged to the household size which is about less or equal to 5 members. A Pearson Chi-square statistic (χ^2^ = 0.000) indicates that there is a statistically significant association between household size and the consumer intention to purchase OGPs. 

Monthly household income positively influences consumers’ decisions to purchase OGPs instead of conventionally grown products [3,52]. A larger proportion (about 48%) of the interviewed consumers’ households in Shelly Centre indicated that they had a monthly income level of between US$ 793.06–US$ 2162.90 (Table 2). Cross-tabulation between the monthly household income categories and the intention to purchase OGPs indicates that a substantial contribution of consumers who agreed to purchase OGPs belonged to the consumers that had a monthly income level of between US$ 793.06–US$ 2162.90. A Pearson Chi-square statistic (χ^2^ = 0.000) indicates that there is a statistically significant association between monthly household income and the consumer intention to purchase OGPs.

### 3.2. Knowledge by the Interviewed Consumers of Shelly Centre about Organically Grown Products

Knowledge about OGPs is very crucial to making purchasing decisions by consumers [17,19,55,56]. The Interviewed consumers in Shelly Centre were asked to indicate if they had any knowledge about OGPs. Table 3 shows that the majority (96%) of the interviewed consumers knew about OGPs. This finding may be because consumers that participated in the study were reasonably educated with at least 12 schooling years (had attained at least matric) (see Section 3.1). This finding suggests that the interviewed consumers in Shelly Centre have basic knowledge about the specific products they consume. This finding is in line with the study of Mhlophe [3], who reveals that the relevance of education is primarily vital in acquiring information and therefore equally important in shaping positive purchase intentions of consumers.

### 3.3. Reliability Analysis

The measure of internal consistency of a questionnaire is crucial since it helps to find how closely related a set of items are as a group. Internal consistency describes the degree to which all items in an investigation will measure a comparable variable and will, therefore, link to the affinity of the rest of the items in that investigation [57]. Usually, a Cronbach’s alpha (α) value of more than 0.70 and closer to 1 is deemed to be a reliable score (acceptable). The Cronbach’s alpha (α = 0.692) value for this dataset used in this paper is slightly lower than 0.70 (Table 4). However, there are provisions that this value is also acceptable.

### 3.4. What Do Organically Grown Products Mean to the Interviewed Consumers in Shelly Centre?

The basic definition of an OGP is that its production uses fewer chemicals or harmful fertilisers [3,58]. Different people may have different opinions about OGPs since they are perceived to have many different attributes. The interviewed consumers in Shelly Centre were asked to indicate their understanding or view what OGPs meant to them. The results in Table 5 show that a higher proportion (about 83%) (that is those who agreed (39.3%) and strongly agreed (44%)) of the interviewed consumers in Shelly Centre are of the view that OGPs are not genetically modified organisms. Overall, the results have a mean agreement/disagreement score of 4.11 (which is close to 4 (that agrees)), suggesting that consumers in Shelly Centre understood that OGPs are not modified to be immune to specific environmental conditions. Studies of Bazzani et al. [4]; Hilverda et al. [5], and Miranda-de la Lama et al. [6], indicate that there is a prohibition to use genetic engineering in the OGPs regulations. The finding implies that the interviewed consumers in Shelly Centre are likely to purchase OGPs with confidence since they believe that OGPs are not genetically modified.

Nowadays, many people are increasingly becoming worried about the use of technology in food production which resultantly damages the environment during production. Williams [59], also mention that the production of OGPs results in fewer pesticide residues in food products, which reduces negative externalities. The results of this paper show that a higher proportion (about 64%) (that is those who agreed (about 31%) and strongly agreed (about 32%)) of the interviewed consumers in Shelly Centre concur to the assertion that OGPs avoids the use of technology (Table 5). Overall, the result has a mean agreement/disagreement score of 3.65 (which is between 3 (neutral) and leaning to 4 (that agrees)), suggesting that the interviewed consumers in Shelly Centre are either indifferent or agree that OGPs production is without the use of technology. 

Suh [58], and Kristiansen et al. [60] reveals that organic agricultural methods are of the natural form and limit the use of harmful chemicals and inorganic fertilisers in their production. Therefore, the production of OGPs combines traditional and modern techniques to conserve the public environment while using it to good use. This method of production is likely to attract environmentally conscious consumers to purchase OGPs. The results of this paper show that a higher proportion (about 79%) (that is those who agreed (about 43%) and strongly agreed (about 36%)) of the interviewed consumers in Shelly Centre believed that OGPs are naturally grown food products (Table 5). Overall, the results have a mean agreement/disagreement score of 3.99 (which is equal to 4 (that agrees)), suggesting that the interviewed consumers in Shelly Centre concur with the idea that OGPs are naturally grown food products. 

### 3.5. How the Interviewed Consumers of Shelly Centre Identify Organically Grown Products on the Market?

The interviewed consumers in Shelly Centre were asked to indicate whether they were able to identify or differentiate OGPs from non-organically grown products. Table 6 shows that a higher proportion (52%) of the interviewed consumers agreed (but not so confident) that they were able to identify or differentiate organically grown products from conventionally grown products. Overall, a mean agreement/disagreement score (3.76) on whether or not the interviewed consumers in Shelly Centre were able to differentiate between OGPs from conventionally grown products suggests a fairly mixed finding (orbiting around the centre ranking score; 3 (that is being neutral), but leaning to the agreement/disagreement score of 4 (that is agree)). 

Concerning the ways how the interviewed consumers in Shelly Centre identified OGPs on the market; the results show that about 39 per cent indicated that they were able to identify OGPs through the physical appearance of the product (Table 6). The remaining higher proportion (about 61%) could not (about 34%) or were not sure (about 32%). Nonetheless, a mean agreement/disagreement score of 2.95 (close to 3 (that is neutral)), suggests the interviewed consumers in Shelly Centre generally were uncertain of whether or not they were able to identify or differentiate OGPs from competing products by physical appearance. This finding confirms the studies of Grzybowska-Brzezinska et al. [61], and Pearson et al. [62], that a majority of consumers are unable to recognise OGPs with a naked eye, but, only through tags (stickers). 

De Villiers [1], and Kisaka-Lwayo and Obi [51], reveal that the certification of OGPs through accredited stickers ordinarily focuses on specific standards and values which are mostly used to create meaning and give assurance about these products. Organic farming is still in the growing phase in South Africa, and in most cases, consumers would purchase OGPs that have accredited labels. The results in Table 6 show that a higher proportion (about 93%) of the interviewed consumers in Shelly Centre (that is about 53 per cent that agreed and 40 per cent that strongly agreed) identified or distinguished OGPs from other products only when they have accredited labels. Overall, the identification of OGPs by accreditation stickers has a mean agreement/disagreement score of 4.33 (that is close to 4 but above 4—leaning to agree (5) strongly), and this suggests that consumers are confident to purchase OGPs when they have accredited labels. Indeed, Bauer et al. [63], and Curtisa et al. [64] reveal that consumers mostly purchase OGPs when they see accredited labels or when sold in approved retail outlets.

### 3.6. The Frequency of Purchasing Organically Grown Products by the Interviewed Consumers in Shelly Centre

Consumers differ in their OGP purchasing frequencies since the products have different values to different consumers [62]. The interviewed consumers in Shelly Centre were asked to indicate how often they purchase OGPs. Results in Table 7 show that a higher proportion (about 41%) of the interviewed consumers in Shelly Centre purchase OGPs once every two weeks (fortnightly), followed by those who purchase OGPs once a month (about 27%). This result is not a surprise given that OGPs are relatively expensive compared to other food products and the general assumption is that consumers purchase OGPs when they have money (when they get paid). The purchase frequency exhibited in these results aligns with the household income levels and earning dates [65]. A higher proportion of people in South Africa usually receive their salaries fortnightly and monthly.

### 3.7. Consumers’ Willingness to Purchase Organically Grown Products Regardless of Price in Shelly Centre

The principal factor on the ever-increasing production of OGPs is the willingness of consumers to pay for them [66]. However, not all consumers can afford OGPs since the desire to spend on a product depends on the ability to pay for it [67]. The interviewed consumers in Shelly Centre were asked to indicate their willingness to purchase OGPs regardless of price; in other words, that is whether they would continue to purchase OGPs should the price increase. The results in Table 8 show that a higher proportion (about 73%) of interviewed consumers in Shelly Centre (that is those that agreed (about 45%) and strongly agreed (28%)) indicates that they would continue to purchase OGPs even if the price continues to rise. Overall, a mean agreement/disagreement score of 3.77 (orbiting around 3 (neutral) but close to 4—leaning to agree), suggests that a fair number of the interviewed consumers in Shelly Beach Centre would be willing to purchase OGPs regardless of price. Consumers may express their willingness to pay a premium price of OGPs especially when they may perceive the products to be healthy and environmentally friendly. As for the reasons apart from the price factor, as to why the consumers of Shelly Beach would purchase OGPs are explored further in Section 3.9 and Section 3.10.

### 3.8. Consumer Perceptions towards Organically Grown Products in Shelly Centre

The finding of this paper show mixed feelings concerning the perceived price of OGPs. About 41 per cent of the interviewed consumers in Shelly Centre believed that OGPs were not expensive (that is about 29 and 12% consumers that disagreed and strongly disagreed respectively), and about 39 per cent were neutral (Table 9). Moreover, the results show a fair level of agreement concerning the perceived price of OGPs (Mean agreement/disagreement = 2.75). The results show that the interviewed consumers in Shelly Centre consider OGPs as being environmentally friendly as releveled by the majority (92%) of the respondents (that is about 75 and 17% that agreed and strongly agreed respectively) (Table 9). Respondents’ level of agreement of the perception that OGPs are environmentally friendly was above average, with a mean agreement/disagreement value of 4.01. Mditshwa et al. [2], and Prada et al. [7] states that one of the most important reasons for the purchase of OGPs by consumers is concerns about food safety. The interviewed consumers who strongly agreed (were confident) that OGPs are safe accounted for about 53 per cent of the sample, and those who agreed (believed but not so confident) accounted for about 43 per cent (Table 9). Moreover, the mean agreement/disagreement score of the perception by the interviewed consumers that OGPs are safe was above average (4.45). About 39 per cent (Table 9) of the interviewed consumers in the Shelly Centre were neutral when asked about the OGPs’ appeal to nature (whether OGPs have a better smell than conventional food products). About 40 per cent (that is 32, and 8 per cent agreed and strongly agreed respectively) that OGPs were appealing to nature concerning their scent (Table 9). The majority (88%) of the interviewed consumers in Shelly Centre (that is 76, and 12% agreed and strongly agreed respectively) that OGPs are high-quality products and have a much better taste when compared to conventionally grown products (Table 9). Table 9 shows that a higher proportion (about 73%) of the interviewed consumers in Shelly Centre agreed that OGPs were generally difficult to access on the market. A higher proportion (88%) of the interviewed consumers in Shelly Centre were of the view that OGPs were only limited to certain retail outlets such as Pick ‘n’ Pay, Spar and Woolworths. Most (96%) (that is 48% agreed and strongly agreed respectively) of the interviewed consumers in Shelly Centre were of the view that OGPs are healthier than conventional food products (Table 9). Again, respondents’ level of agreement was above average for the perceptions that OGPs have a better smell, have a good taste and of high quality, are difficult to access on the market, and are healthy which had a mean agreement/disagreement values of 3.27, 3.92, 3.49 and 4.36 respectively (Table 9).

### 3.9. Correlation Pearson Analysis

Pearson’s correlation coefficient measures the strength of a linear relationship between two variables. The dependent variable in this analysis is the consumer’s intention to purchase OGPs. The correlation Pearson analysis here tests for statistical association between the dependent variable and the independent (explanatory) variables. The explanatory variables in this analysis include the demographic characteristics and consumers’ perceptions towards OGPs. The demographic characteristics considered include the age group, gender, ethnicity, marital status, employment status, household size, and monthly household income of the consumer. Concerning consumer perceptions towards OGPs; they include beliefs that; OGPs are highly priced, environmentally friendly, safe, have a better smell, have a good taste and of high quality, difficult to find on the market, and healthy. The correlation Pearson analysis results in Table 10 reveal that demographics variables; ethnicity (r-value of 0.182 at 95% confidence interval), and monthly household income (r-value of 0.287 at 99% confidence interval) have a statistically significant association with the consumer purchase intentions of OGPs. Concerning consumer perceptions; beliefs that OGPs are highly priced (r-value of −0.274 at 99% confidence interval), environmentally friendly (r-value of 0.586 at 99% confidence interval), are safe (r-value of 0.525 at 99% confidence interval), have a better taste and of high quality (r-value of −0.239 at 99% confidence interval), are difficult to find on the market (r-value of −0.215 at 99% confidence interval), and are healthy (r-value of 0.628 at 99% confidence interval) have a statistically significant association with the consumer purchase intentions of OGPs. On the other hand, the education level of the consumer, age group, gender, marital status, household size, employment status ant the perception that OGPs have a better smell does not show a statistically significant association with the consumer purchase intentions of OGPs in this dataset (Table 10). Only those variables that show a statistically significant association with the dependent variable in the correlation Pearson analysis are interrogated further in the multiple regression analysis formulated and specified in Section 2.4.

### 3.10. Factors Influencing Consumer Purchase Intentions of Organically Grown Products in Shelly Centre

A multiple regression analysis was used to test for the factors that influence the purchase intentions of OGPs in Shelly Centre in Port Shepstone. The dependent variable used in this study is the consumer purchase intention or decision. The independent variables inputted in the multiple regression model include the ethnicity of the consumer, monthly household income, and consumer perceptions towards the purchase of OGPs that include: OGPs are highly priced, environmentally friendly, are safe, have a better taste, and of high quality, are difficult to find on the market, and healthy. The goodness-of-fit of the model computed for this dataset included the R-Square, F-statistic and the variance inflation factors (VIF) statistics. Out of the eight (8) variables built-in the multiple regression model, five (5) remain in the final model after the collinearity diagnostics. Only those variables with a VIF of less than four (4) remain in the final model with the following variables dropped due to multicollinearity; OGPs are environmentally friendly, safe, and healthy. The R-Square and the adjusted R-Square statistic value for this dataset is 0.4705 and 0.4404 respectively (Table 11). The F test results show that the F statistic, F (8, 141) is 15.66 and the *p*-value associated with the F statistic, F > prob is *p* = 0.000 (Table 11). Based on the goodness-of-fit of the model results, the null hypothesis is rejected with extremely high confidence—above 99.99%, implying the model provides a better fit than the intercept-only model. Results in Table 11 show the factors that significantly influence the purchase intentions of OGPs in Shelly Centre. These variables include the ethnicity of the consumer, monthly household income, and consumer perceptions towards OGPs (that is OGPs are highly priced, have a good taste, and of high quality, and their difficulty to find OGPs on the market).

The variable *ethnicity*
*of the consumer*, inputted in the multiple regression model as a dummy variable (the ethnic group “African” takes the value 0 and “not of African descendant” (Coloured, White or Asian) takes the value 1). Table 4 shows that ethnicity of the consumer is statistically significant at 1% level (*p* = 0.000), and belonging to the ethnic group, not of African descendant (Coloured, White or Asian) is positively correlated with the consumer purchase intention of OGPs). This finding is consistent with the expected outcome.

The variable *monthly household income*, included in the multiple regression model consists of five categories or income constructs “less than US$ 720.97; US$ 793.06–US$ 2162.90; US$ 2235–US$ 2883.87; US$ 2955.97–US$ 3604.84 and more than US$ 3604.84”. Set as a reference or base here is the income group of consumers with less than US$ 720.97 per month. The results in Table 5 show a statistically significant positive relationship between the income groups and the purchase intentions of OGPs when compared to the income base category. This finding is consistent with the expected outcome.

The *perception that OGPs are of high price* variable is found to be statistically significant at 1% level (*p* = 0.000) and negatively correlated with the purchase intentions of OGPs. This finding is consistent with the expected outcome.

The *perception that OGPs have a better taste and of a high-quality* variable is found to be statistically significant at 5% level (*p* = 0.004) and negatively correlated with the purchase intentions of OGPs. This finding is an unexpected one and in contrast with the expected outcome.

The *perception*
*that OGPs are difficult to find on the market* is found to be statistically significant at 1% level (*p* = 0.000) and negatively correlated with the purchase intentions of OGPs. This finding is consistent with the expected outcome. 

## 4. Discussion

The purpose of this paper is to identify and assess the factors that influence consumer purchase intentions of OGPs in Shelly Centre, in Port Shepstone. The finding of this paper show that most of the interviewed consumers in Shelly Centre generally had attained some education with a minimum education level being a matric qualification and an average of 15 schooling years. This finding suggests that on average, the education level of the interviewed consumers in Shelly Centre was a junior undergraduate degree. Dettmann and Dimitri [45], indicates that consumers with a higher education level are more likely to have positive purchase intentions of OGPs compared to consumers with less or no education. Nonetheless, no statistically significant association is detected in this paper between the level of education and the consumer purchase intentions of OGPs in Shelly Centre.

Most of the interviewed consumers in Shelly Centre belonged to the economically active group of the population. A cross-tabulation statistic shows that a substantial proportion of consumers who had a definite intention to purchase OGPs in Shelly Centre are in the age group of between 35 and 49 years. This finding suggests that the purchase of OGPs is mostly by the economically active members since they may afford the price premium associated with them. This finding is in line with those of Mhlophe [3], and Engel [12], that many people ordinarily become active consumers of OGPs between the ages of 26 and 35 years. Nonetheless, no statistically significant association is detected in this paper between the age of the consumer the consumer purchase intentions of OGPs in Shelly Centre.

The interviewed consumers in Shelly Centre were mainly females. A cross-tabulation statistic shows that a substantial proportion of consumers who had an affirmative intention to purchase OGPs in Shelly Centre are females, reinforcing the notion that women are the dominant purchasers of OGPs more than men. This finding could be due to that women are more concerned about their health and want to live a quality lifestyle [46]. However, no statistically significant association is detected in this paper between the gender of the consumer the consumer purchase intentions of OGPs in Shelly Centre. 

The ethnic group; not of African descent (Coloured, White or Asian) mainly dominated the interviewed consumers in Shelly Centre. A cross-tabulation statistic shows that a substantial proportion of consumers who belong to the ethnic group; not of African descent (Coloured, White or Asian) had a definite intention to purchase OGPs in Shelly Centre. This finding suggests that the ethnic group; not of African descent (Coloured, White or Asian) were the primary consumers of fruit and vegetable OGPs in Shelly Centre. This finding is not a surprise that the ethnic group; not of African descent (Coloured, White or Asian) consumers would dominate the OGP market since their income levels are much higher in South Africa compared to the African ethnic group [68]. The privilege is thought to enable consumers who are not of African descent to afford purchasing OGPs regardless of the price premium attached to it [69]. 

Further analysis in the multiple regression model shows that the ethnic group is statistically significant with the consumer purchase intentions of OGPs. The multiple regression model predicts that consumers who are not of African descent (Coloured, White or Asian) ethnic group are more likely to have positive purchase intentions towards OGPs than those consumers that belong to the African ethnic group. Stats SA [70], reveal that people who are not of African descent (Coloured, White or Asian) ethnic group share a universal cultural system towards organic products. Therefore, this is assumed to be the reason the consumers who are not of African descent (Coloured, White or Asian) are more likely to have a positive purchase intention of OGPs than consumers of African descent.

The finding in this paper reveals that the interviewed consumers in Shelly Centre were mostly single, divorced or widowed. A cross-tabulation statistic shows that a relatively higher proportion of consumers who are single (unmarried) had a definite intention to purchase OGPs. This finding suggests that single or unmarried consumers purchase OGPs more often than their counterparts (married consumers). This finding is in agreement to studies for example by Hawk [71], and Joifin [72], that single consumers spend conspicuously more on food as compared to married consumers would do. Nonetheless, no statistically significant association is detected in this paper between the marital status of the consumer and the consumer purchase intentions of OGPs in Shelly Centre.

The majority of the interviewed consumers of OGPs in Shelly Centre were found to be full time employed. A cross-tabulation statistic shows that a substantial proportion of consumers that had a definite intention to purchase OGPs are employed. This finding implies that employed consumers have a stable income and may worry about the quality of food or products they eat in addition to the ability to be food secure. This finding is also in line with Basha et al. [8] that people that are employed can afford products of high quality compared to unemployed people. Nonetheless, no statistically significant association is detected in this paper between the employment status of the consumer and the consumer purchase intentions of OGPs in Shelly Centre. 

A reasonably higher proportion of the interviewed consumers in Shelly Centre belonged to the household size which is about less or equal to five members. A cross-tabulation statistic further shows that consumers who reside in a small household (family) size had a definite intention to purchase OGPs in Shelly Centre. The finding suggests that consumers from smaller household (family) size may be willing to purchase OGPs than consumers from larger household (family) size, especially considering the costs (price premium) attached with the consumption of OGPs. This finding is in line with Mhlophe [3], and Slamet et al. [53], that consumers with small household (family) size are more willing to purchase OGPs than those consumers from households with larger family size. A Pearson Chi-square statistic shows a statistically significant association between household (family) size and the intention to purchase OGPs. However, further interrogation (correlation Pearson analysis) detects no statistically significant association of this variable with the consumer purchase intentions of OGPs in Shelly Centre.

The finding in this paper from a cross-tabulation statistic is that a higher proportion of consumers receiving monthly household income of between US$ 793.06–US$ 2162.90 show a definite intention to purchase OGPs. Consumers in the income category of between US$ 793.06–US$ 2162.90 suggests that the interviewed consumers in Shelly Centre have a reasonable monthly household income which may enable them to purchase OGPs regardless of the price premium attached to it. Again, results earlier on show that a higher proportion of the interviewed consumers in Shelly Centre have a reasonable education. Mhlophe [3], reveals that consumers with a somewhat low-income level but well informed and educated are also likely to purchase OGPs. The multiple regression model shows a statistically significant and positive correlation between monthly household income and the intention to purchase OGPs. The multiple regression model analysis predicts that the magnitude to purchase OGPs increases with a higher income category of the consumer for example consumers in the income category of between US$ 2955.97–US$ 3604.84 and earning more than US$ 3604.84, the magnitude would increase by 0.68 and 1.26 respectively per unit increase in income than those consumers in the lower income category (base category) (Table 11). The finding is in harmony with studies of Mhlophe [3], and Slamet et al. [53], which states that a high monthly household income positively influences consumers’ decisions to purchase OGPs rather than conventionally grown products.

Concerning consumer perceptions towards purchase intentions of OGPs, the finding is a mixed feeling that OGPs are highly priced. A somewhat higher proportion of the interviewed consumers in Shelly Centre did not conform to the view that OGPs are highly priced but also a fair portion was somewhat in-between or agreed to this assertion. The interviewed consumers in Shelly Centre were asked to indicate their willingness to purchase OGPs should the price increase. The finding is somewhat of mixed feelings with a fair number of consumers reporting that they would continue to purchase OGPs regardless of price and yet some uncertain. Nonetheless, the multiple regression model analysis shows that the perception by consumers that OGPs are of a higher price is statistically significant and negatively correlated with the consumer purchase intentions of OGPs. The multiple regression model analysis predicts that a consumer who perceives that OGPs are highly priced compared to conventional food products is less likely to purchase OGPs by about 14 per cent. This finding agrees with studies by Lee and Yun [23], and Marian et al. [47], that show that a higher price associated with OGPs would lessen their purchase. 

Lee and Yun [23], Ling [24], and Chang and Chang [25], asserts that one of the main reasons for the increase in the demand of OGPs is that many consumers have taken an initiative to protect the environment (that is being environmentally conscious). The finding in this paper show that most of the interviewed consumers in Shelly Centre consider OGPs as being environmentally friendly. A statistically significant association is detected in this paper (correlation Pearson analysis) between the consumer perception that OGPs are environmentally friendly and the intention to purchase OGPs. However, the inclusion of this variable for further investigation in the multiple regression model was not possible due to multicollinearity. 

Mditshwa et al. [2], and Prada et al. [7] states that one of the most important reasons for the purchase of OGPs by consumers is concerns about food safety. However, McFadden & Huffman [73], state that the articulation of “food safety” theory is not clear. Lack of clarity on the “food safety” concept by consumers, therefore, can make it difficult to make definitive decisions by consumers about OGPs being safe when compared to other products. Generally, the interviewed consumers in Shelly Centre were confident that OGPs are safe. A statistically significant association is detected in this paper (correlation Pearson analysis) between the consumer perception that OGPs are safe and the intention to purchase OGPs. However, the inclusion of this variable for further investigation in the multiple regression model was not possible due to multicollinearity. 

The appealing nature of OGPs proxied in this paper by the consumer perception that OGPs have a better smell has been one of the most critical reported features in attracting consumers’ eyes into purchasing them [74]. However, there were some mixed feelings by the interviewed fruit and vegetable consumers of OGPs in Shelly Centre concerning the perception that OGPs have a better smell. This finding suggests that some consumers purchased OGPs based on the perceived product appeal to nature, while, other features could have interested other consumers. Nonetheless, a mean value higher than the average suggests that the consumers in Shelly Centre confirmed the assertion that OGPs have a better appeal to nature concerning their scent compared to conventional food products. Nonetheless, there is no statistically significant association detected in this paper (correlation Pearson analysis) between the consumer perception that OGPs have a better smell and the intention to purchase OGPs. Hence, the exclusion of this variable for further interrogation of its influence on the consumer purchase intentions of OGPs in the multiple regression model. 

The majority of the interviewed consumers in Shelly Centre believed that OGPs are high-quality products and have a much better taste compared to conventionally grown products. This finding is in line with Mditshwa et al. [2]; Marian et al. [47]; Bryła [49], and Thøgersen et al. [75], that taste and product quality is one of the many essential factors that influence the purchase of OGPs. The perception that OGPs have a good taste and to be of high quality was found to be statistically significant in influencing consumer purchase intentions. Paradoxically, this perception that OGPs have good taste and to be of high quality is found to have a negative correlation with the consumer purchase intentions of OGPs in Shelly Centre. The multiple regression model analysis predicts that the perception by consumers that OGPs have a better taste and to be of high quality is more likely to decrease a consumer’s purchase intention of OGPs by about 51 per cent (Table 11). This finding is surprising and in contrast with the several studies, for example, Mditshwa et al. [2]; Marian et al. [47]; Bryła [49]; Thøgersen et al. [75], which postulate that the attribute such as product better taste and perceived high quality would increase the purchase intentions of OGPs. The finding could be explainable in the sense that this claim that OGPs have a better taste and to be of high quality is not justifiable to all OGPs as revealed by Mukul et al. [74], and Dumea [76]. Another reason for this conflicting finding could be the fact that generally OGPs are priced higher than other conventional food products. This justification is evident from the correlation Person analysis where the perception by consumers that OGPs have a better taste and to be of high quality is statistically significant and negatively correlated with the perception that OGPs are generally highly priced (Table 10). Therefore, consumers may see this claim that OGPs have a better taste and to be of high quality as a marketing strategy to inflate the price of OGPs. This reasoning may eventually influence consumers to stick to purchasing cheaper conventional food products, even though they may believe OGPs to have a better taste and to be high-quality products. 

Additionally, the negative correlation could be due to the other food quality attributes. For example, food quality acknowledgement can be through branding, packaging, and nutritional information. In as much as consumers perceive OGPs to be of better taste and quality, consumers may not be inclined to purchase OGPs due to limited branding and certification. A higher proportion of the interviewed consumers in Shelly Centre attested that they are only able to identify or distinguish OGPs only through accredited stickers or accreditation. Full accreditation or certification of OGPs may not be the case especially in South Africa; where organic farming is relatively new and still a growing sector. However, this finding calls for further research to unpack the intricacy of it.

Generally, the feeling by the interviewed consumers in Shelly Centre is that OGPs are difficult to find on the market. Most of the interviewed consumers in Shelly Centre indicated that OGPs marketing is limited only to specific retail outlets such as Pick ‘n’ Pay, Spar and Woolworths. This finding suggests that “indeed” OGPs are difficult to access on the market in South Africa and only limited to accredited retail outlets. The multiple regression model analysis shows that the perception that OGPs are difficult to find on the market is statistically significant and negatively correlated with the consumer purchase intention of OGPs. The multiple regression model analysis predicts that a consumer who perceives that OGPs are difficult to access on the market is likely to decrease his/her purchase intention of OGPs by about 41 per cent (Table 11). The finding implies that limited availability of OGPs on the market decreases their accessibility and thus lessens the purchase intentions of OGPs by consumers. The finding agrees with the studies of Kisaka-Lwayo and Obi [51]; Yao and Kaval [77], and Ndungu [78], who state that OGPs are generally difficult to access on the market as they are not readily available.

Nowadays, consumers have become more concerned about their health, and some have directed their attention to OGPs since their consumption is primarily seen to be healthy [8,52,59,79]. Generally, most of the interviewed consumers in Shelly Centre were in agreement with the assertion that OGPs are healthier than conventional food products. A statistically significant association is detected in this paper (correlation Pearson analysis) between the consumer perception that OGPs are healthier and the intention to purchase OGPs in Shelly Centre. Nonetheless, further investigation in the multiple regression model analysis shows that this variable is not statistically significantly influencing the consumer intention to purchase OGPs in Shelly Centre. 

## 5. Conclusions

There are relatively few studies which have been conducted to analyse and determine consumers’ purchase intentions of OGPs in South Africa. The results of this study can be of great importance to both retailers and consumers of OGPs since they provide valuable information on consumer’s purchase intentions. The results also have implications for policy makers and researchers. 

The findings of this paper can be used to inform and educate consumers about OGPs. A statistically significant association between the education status of the consumer (a proxy for consumer knowledge) and the intention to purchase OGPs is not detected in this paper. However, a closer look in the correlation Pearson analysis (Table 10) shows that a positive correlation exists between the income level of the consumer. The income of the consumer is in turn found to be a statistically significant factor in shaping a positive consumer purchase intention of OGPs in this paper. Again, the income of the consumer shows a positive correlation to other crucial cross-cultural factors such as ethnicity which is also a statistically significant factor in shaping a positive consumer purchase intention of OGPs in this paper. The implication here is that consumer education (knowledge) may primarily raise awareness to consumers on numerous attributes associated with the consumption of OGPs and eventually create a positive social change (consumer behaviour towards OGPs). Therefore, this may promote the purchase of OGPs once consumers understand such information. Quality information by consumers about OGPs is not only a good inspiration for social change but good health and the environment.

Concerning managerial implications; generally, OGPs are priced higher than conventional food products. Because consumers require a return value for their money, the perceived high price of OGPs has been one reason for consumers’ failure to develop a positive purchase intention. Arguably so, the price factor conflicts with other motivations for creating a positive purchase intention of OGPs such as environmental and safety concerns. Even in cases where consumers can afford the price premiums, consumers lack the facts to justify the price premiums attached to the intended purchases of OGPs and thus limit the consumers’ purchase intentions of OGPs. For retailers and marketers to encourage a positive purchase intention of OGPs, the benefits of consuming OGPs will have to be communicated clearly and effectively to the consumers. This clarity could be addressed through for example the use of advertisements and branding (labelling) and food certification that can change consumers’ contrary purchase intentions of OGPs for example that these products are ludicrously valued and priced. Although the price of OGPs is perceived as a significant barrier to their purchase by consumers, Rödiger and Hamm [80] suggest that effective design of pricing strategies is still lacking. There is, therefore, a need for further research to focus on the consumer price-sensitive behaviour (price elasticity of demand, and the willingness to pay for OGPs) across different income categories of consumers. It is also evident from the findings that one of the limitations of shaping a positive purchase intention of OGPs by consumers is the difficulty to access the product on the market. The managerial implication here is that; retailers (marketers) need to design strategies and elements (marketing mix) that will ensure that OGPs are readily available to consumers. 

The purchase of OGPs is arguably one way to preserve the environment and maintain the health of individuals and society. The government can have a role to play in promoting the purchase of OGPs in South Africa, which will ultimately impact the well-being of society. Generally, worldwide, the occurrence of illnesses by individuals caused by consuming processed food products having harmful chemical residues is on the rise. Hence the consumption of organic foods has been one of the health and environmental solutions in some countries. In this paper, we found that the price of OGPs is one limiting factor in the purchase of OGPs in Shelly Centre. To promote the purchase of OGPs in South Africa, from a policy perspective, the government can offer support such as a consumer price subsidy to make OGPs affordable. Again, consumers may also not be inclined to purchase OGPs due to limited branding and certification, which may also have linkages to other attributes such as the perceived product quality and mistrust by consumers. One surprising finding in this paper is that the perceived taste and quality of OGPs was statistically significant but negatively influencing their purchase intention by consumers. Therefore, there is a need to build trust among the consumers of OGPs through effective government regulations and certification around the marketing of OGPs.

An increasing number of studies reveal that the reasons advocated by the consumers for purchasing OGPs are different and mostly the motives behind their decision to purchase involve concerns for the environment, health and safety. It is evident from this paper that consumer behaviour concerning purchase intentions of OGPs is not a straight forward issue. Various factors including demographics and consumer perceptions drive consumers’ willingness or deter them from purchasing OGPs. From this paper, demographic characteristics; that is ethnicity (not of African descendant), and household income are found to promote a positive purchase intention of OGPs in Shelly Centre in Port Shepstone. Attributes such as consumer concerns for the environment, health and safety are found to be statistically insignificant. Apart from the belief that OGPs are known for their superior quality and freshness than conventional food products, the consumer perception that OGPs are highly priced and have a better taste and quality is found to promote a negative purchase intention of OGPs in Shelly Centre in Port Shepstone. This finding was unexpected and warranties further research to understand more of the intricacies surrounding the consumer purchase intentions of OGPs in South Africa. This paper, therefore, encourages researchers to undertake further similar research but of extended scope on the attributes influencing consumer purchase intentions of OGPs in South Africa to increase the reliability and efficacy of the current findings. This research can be further expanded by for example increasing the sample size and extending the research to other areas in South Africa to enhance the reliability of the findings. Additionally, future research can incorporate numerous marketing strategies used by OGPs marketers, and other attributes that may influence consumers’ preference of OGPs not included in this paper.

## Figures and Tables

**Figure 1 ijerph-16-00956-f001:**
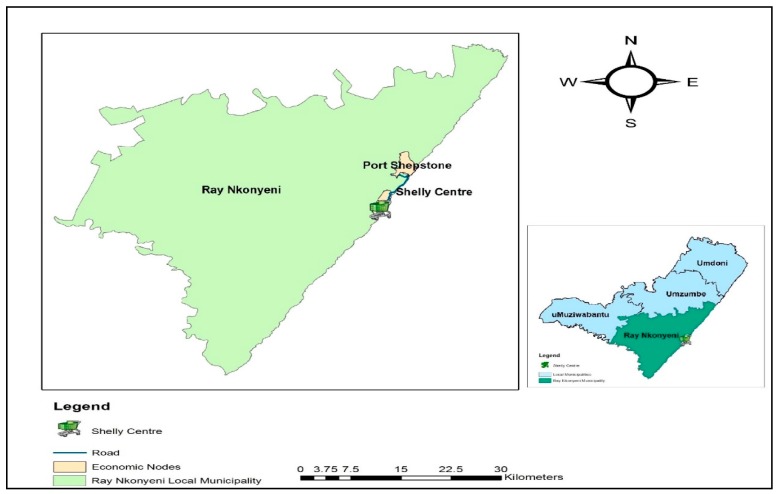
Map showing the location of Shelly Centre in Ray Nkonyeni Local Municipality.

**Figure 2 ijerph-16-00956-f002:**
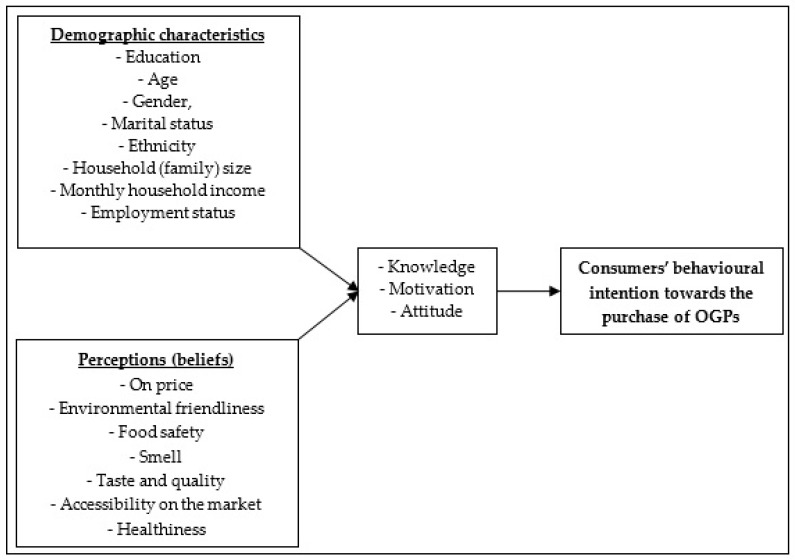
The conceptual framework of the factors influencing consumer purchase intentions of OGPs.

**Table 1 ijerph-16-00956-t001:** Explanatory variables, description and their expected outcome.

Independent/Explanatory Variable	Variable Description	Measurement Type	Expected Outcome (±)
The education level of the consumer	The level of education of the respondent (number of schooling years)	Continuous	+
Age group of consumers	Age groups of the respondent in years (1 = 21–29; 2 = 30–34; 3 = 35–49; 4 = 50–60; 5 = >60	Categorical	+/−
Gender of consumer	Gender of the respondent (0 = male; 1 = female)	Categorical (Dummy)	+/−
Marital status of consumer	Marital status of the respondent (0 = single, divorced or widowed; 1 = married)	Categorical (Dummy)	+/−
Ethnicity of consumer	The ethnicity of the respondent (0 = African; 1 = Not of African descendant (Coloured; White or Asian))	Categorical (Dummy)	+/−
Household size	Actual respondent’s family size (1 = less or equal to 5; 2 = 6–10; 3 = 11–15; 4 = >15)	Categorical	-
Monthly household income	Monthly income (1 = Less than US$ ^1^ 720.97; 2 = US$ 793.06–US$ 2162.90; 3 = US$ 2235–US$ 2883.87; 4 = US$ 2955.97–US$ 3604.84; 5 = More than US$ 3604.84)	Categorical	+
Employment status of the consumer	Employment status of respondent (0 = unemployed; 1 = employed)	Categorical (Dummy)	+
Perception that OGPs are highly priced	Organically grown products are expensive compared to other products (1 = strongly disagree; 2 = disagree; 3 = neutral; 4 = agree; 5 = strongly agree)	Continuous	-
Perception that OGPs are environmentally friendly	Organically grown products remains are beneficial for enriching the environment (1 = strongly disagree; 2 = disagree; 3 = neutral; 4 = agree; 5 = strongly agree)	Continuous	+
Perception that OGPs are safe	Organically grown products are safe compared to other products (1 = strongly disagree; 2 = disagree; 3 = neutral; 4 = agree; 5 = strongly agree)	Continuous	+
Perception that OGPs have a better smell	Organically grown products have a better smell than other conventional foods (1 = strongly disagree; 2 = disagree; 3 = neutral; 4 = agree; 5 = strongly agree)	Continuous	+
Perception that OGPs have a better taste and of high quality	Organically grown products have a better taste and of high quality than other conventional products (1 = strongly disagree; 2 = disagree; 3 = neutral; 4 = agree; 5 = strongly agree)	Continuous	+
Perception that OGPs are difficult to find on the market	Organically grown products are difficult to access on the market (1 = strongly disagree; 2 = disagree; 3 = neutral; 4 = agree; 5 = strongly agree)	Continuous	-
Perception that OGPs are healthy	Organically grown products are healthier compared to other conventional products (1 = strongly disagree; 2 = disagree; 3 = neutral; 4 = agree; 5 = strongly agree)	Continuous	+

^1^ US$: Denotes United States of America Dollar.

**Table 2 ijerph-16-00956-t002:** Demographic characteristics of the interviewed consumers in Shelly Centre (n = 150).

The Education Level of the Consumer						
Minimum 12	Maximum 22	Mean 14.99	Mode 16	Std. Deviation 2.356	Variance 5.550	N 150
**Demographic Characteristic**			**Frequency**		**Percentage (%)**	
**Age group of the consumer (years)**						
21–29			43		28.67	
30–34			42		28	
35–49			53		35.33	
50–60			9		6	
Above 60			3		2	
**Total**			**150**		**100**	
**Gender of the consumer**						
Male			56		37.3	
Female			94		62.7	
**Total**			**150**		**100**	
**The ethnicity of the consumer**						
African			52		34.7	
Not of African descendant (Coloured, White or Asian)			98		65.3	
**Total**			**150**		**100**	
**Marital status of the consumer**						
Single, divorced or widowed			80		53.3	
Married			70		46.7	
**Total**			**150**		**100.0**	
**Employment status of the consumer**						
Unemployed			17		11.3	
Employed			133		88.7	
**Total**			**150**		**100.0**	
**Household size**						
Less or equal to 5			68		45.3	
6–10			64		42.7	
11–15			18		12.0	
**Total**			**150**		**100.0**	
**Monthly household income**						
Less than US$ 720.97			41		27.3	
US$ 793.06–US$ 2162.90			72		48.0	
US$ 2235–US$ 2883.87			13		8.7	
US$ 2955.97–US$ 3604.84			18		12	
More than US$ 3604.84			6		4	
**Total**			**150**		**100**	

**Table 3 ijerph-16-00956-t003:** Knowledge by Consumers of Shelly Centre about organically grown products (n = 150).

Do You Have Any Knowledge about Organically Grown Products?	Frequency	Percentage (%)
No	6	4.0
Yes	144	96.0
**Total**	**150**	**100.0**

**Table 4 ijerph-16-00956-t004:** Results of reliability test (Cronbach alpha) (n = 150).

Cronbach’s Alpha	Cronbach’s Alpha Based on Standardized Items	N of Items
0.692	0.694	15

N denotes number.

**Table 5 ijerph-16-00956-t005:** Meaning of organically grown products as understood by interviewed consumers in Shelly Centre (n = 150).

Meaning of an OGP as Understood by the Interviewed Consumers in Shelly Centre	Level of Agreement/Disagreement (%)						
	SD (1)	D (2)	N (3)	A (4)	SA (5)	$\bar{x}$	σ
Organically grown products are not genetically modified organisms	8.0	-	8.7	39.3	44.0	4.11	1.114
Organically grown products production avoids the use of technology	12.0	8.0	16.0	31.3	32.7	3.65	1.332
Organically grown products are naturally grown food products	8.0	-	12.7	43.3	36.0	3.99	1.102

SD; D; N; A; SA; $\bar{x}$; σ denotes strongly disagree; disagree; neutral; agree; strongly agree; sample mean & standard deviation.

**Table 6 ijerph-16-00956-t006:** Identification of OGPs by the interviewed consumers in Shelly Centre (n = 150).

Identification of OGPs by Consumers in Shelly Centre	Level of Agreement/Disagreement (%)						
	SD (1)	D (2)	N (3)	A (4)	SA (5)	$\bar{x}$	σ
Are you able to identify or differentiate organically grown products from non-organically grown products?	4.0	-	28.0	52.0	16.0	3.76	0.865
I can identify organically grown products through physical appearance	16.0	12.7	32.0	39.3	-	2.95	1.079
I identify organically grown products through accreditation stickers	-	-	7.3	52.7	40.0	4.33	0.608

SD; D; N; A; SA; $\bar{x}$; σ denotes strongly disagree; disagree; neutral; agree; strongly agree; sample mean & standard deviation.

**Table 7 ijerph-16-00956-t007:** The frequency of purchasing organically grown products by the interviewed consumers in Shelly Centre (n = 150).

How Often Do You Purchase Organically Grown Products?	Frequency	Percentage (%)
Daily	12	8.0
Once a week	36	24.0
Once every two weeks (fortnightly)	62	41.3
Once a month	40	26.7
**Total**	**150**	**100.0**

**Table 8 ijerph-16-00956-t008:** Consumers’ willingness to purchase organically grown products regardless of price in Shelly Centre (n = 150).

Willingness to Purchase Organically Grown Products by Consumers in Shelly Centre	Level of Agreement/Disagreement (%)						
	SD (1)	D (2)	N (3)	A (4)	SA (5)	$\bar{x}$	σ
If the price of organically grown products continue to rise, would you still be willing to buy them?	7.3	8.7	11.3	44.7	28.0	3.77	1.165

SD; D; N; A; SA; $\bar{x}$; σ denotes strongly disagree; disagree; neutral; agree; strongly agree; sample mean & standard deviation.

**Table 9 ijerph-16-00956-t009:** Consumer perceptions towards organically grown products in Shelly Centre (n = 150).

Consumer Perceptions towards OGPs	Level of Agreement/Disagreement (%)						
	SD (1)	D (2)	N (3)	A (4)	SA (5)	$\bar{x}$	σ
Organically grown products are highly priced	12.0	28.7	39.3	12.0	8.0	2.75	1.074
Organically grown products are environmentally friendly	4.0	-	4.0	75.3	16.7	4.01	0.755
Organically grown products are safe	-	4.0	-	43.3	52.7	4.45	0.700
Organically grown products have a better smell	-	20.7	39.3	32.0	8.0	3.27	0.882
Organically grown products have a good taste and of high quality	4.0	-	8.0	76.0	12.0	3.92	0.747
Organically grown products are difficult to find on the market	4.0	16.0	7.3	72.7	-	3.49	0.903
Organically grown products are healthy	4.0	-	-	48.0	48.0	4.36	0.846

SD; D; N; A; SA; $\bar{x}$; σ denotes strongly disagree; disagree; neutral; agree; strongly agree; sample mean & standard deviation.

**Table 10 ijerph-16-00956-t010:** Correlation Pearson analysis results between the independent variables and the consumer purchase intentions of OGPs (n = 150).

	Intention to Purchase OGPs (DV)	Education Level	Age Group	Gender	Marital Status	Ethnicity	Household Size	Monthly Household Income	Employment Status	OGPs Are Highly Priced	OGPs are Environmentally Friendly	OGPs Are Safe	OGPs Have a Better Smell	OGPs Have a Better Taste and of High Quality	OGPs Are Difficult to Find on the Market	OGPs Are Healthy
Intention to purchase OGPs (DV)	1															
Education level	0.024	1														
Age group	0.076	0.244 **	1													
Gender	−0.040	0.067	0.238 **	1												
Marital status	−0.079	0.116	0.375 **	0.176 *	1											
Ethnicity	0.182 *	0.016	0.216 **	0.110	0.344 **	1										
Household size	−0.050	0.154	−0.081	0.122	−0.141	−0.062	1									
Monthly household income	0.287 **	0.314 **	0.348 **	−0.210 **	0.373 **	0.280 **	−0.361 **	1								
Employment status	−0.174	0.287 **	0.374 **	0.195 *	0.345 **	0.033	−0.148	0.170 *	1							
OGPs are highly priced	−0.274 **	0.039	−0.120	0.060	−0.057	0.037	0.123	−0.357 **	−0.231 **	1						
OGPs are environmentally friendly	0.586 **	−0.034	0.359 **	0.007	0.382 **	0.360 **	0.070	0.447 **	0.003	−0.104	1					
OGPs are safe	0.525 **	0.038	0.282 **	0.129	0.254 **	0.312 **	0.214 **	0.111	0.034	−0.115	0.768 **	1				
OGPs have a better smell	−0.003	0.137	0.050	−0.242 **	0.295 **	0.387 **	−0.136	0.388 **	−0.101	0.341 **	0.254 **	−0.085	1			
OGPs have a better taste and of high quality	−0.239 **	−0.131	0.328 **	0.224 **	0.137	0.104	−0.062	0.232 **	0.347 **	−0.263 **	0.069	−0.027	0.123	1		
OGPs are difficult to find on the market	−0.215 **	0.008	−0.055	−0.218 **	0.377 **	0.226 **	−0.222 **	0.190 *	0.210 **	0.143	0.004	−0.197 *	0.338 **	−0.290 **	1	
OGPs are healthy	0.628 **	−0.039	0.284 **	0.176 *	0.219 **	0.344 **	0.112	0.168 *	0.036	−0.265 **	0.826 **	0.889 **	0.035	0.176 *	−0.261 **	1

DV; *; ** denotes dependent variable; and correlation is significant at the 0.05 and 0.01 level (2-tailed).

**Table 11 ijerph-16-00956-t011:** Results of the multiple regression model analysis on factors influencing the purchase intentions of organically grown products in Shelly Centre.

Parameter	Coef.	Std. Err.	t	*p* > |t|	[95% Conf. Interval]		VIF
Constant	7.270368	0.3778446	19.24	0.000	6.523395	8.017341	-
The ethnicity of consumer_Not of African descendant (Coloured, White or Asian)	0.4356981 ***	0.1034605	4.21	0.000	0.2311637	0.6402324	1.23
Monthly household income_Less than US$ 720.97							
US$ 793.06–US$ 2162.90	0.3944465 **	0.1300731	3.03	0.003	0.1373009	0.651592	2.14
US$ 2235–US$ 2883.87	0.2160102	0.1976637	1.09	0.276	−0.1747574	0.6067778	1.57
US$ 2955.97–US$ 3604.84	0.6892454 ***	0.1747316	3.94	0.000	0.343813	1.034678	1.64
More than US$ 3604.84	1.260846 ***	0.2610211	4.83	0.000	0.7448251	1.776867	1.33
The perception that OGPs are highly priced	−0.1425993 **	0.0493452	−2.89	0.004	−0.2401514	−0.0450473	1.46
The perception that OGPs have a better taste and of high quality	−0.5086212 ***	0.0661413	−7.69	0.000	−0.639378	−0.3778643	1.35
The perception that OGPs are difficult to find on the market	−0.4118071 ***	0.0587184	−7.01	0.000	−0.5278894	−0.2957248	1.41
Number of observations	150						
F (8, 141)	15.66						
Prob > F	0.0000						
R-squared	0.4705						
Adj R-squared	0.4404						

***; ** denotes statistically significant at 1% & 5% levels.

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
