# Peer review of "Factors Influencing Consumer Purchase Intentions of Organically Grown Products in Shelly Centre, Port Shepstone, South Africa"

_ijerph, 2019, doi:10.3390/ijerph16060956_

Round 1

Reviewer 1 Report

Dear authors,

The paper is really interesting. The subject you analyze is new an has really social interest. The empirical methodology is good, and the results and conclusions offered are really interested. But I miss  a theoretical section where you offer a deep explanation of your model, I mean, I think you should justify why you choose these variables, how the can be defined and what are the relations among them. As academic, I can not understand a scientific paper without a theoretical section where the model used would be explained and defended. 

Author Response

.

Reviewer 2 Report

The study is promising since there is not much information in the existing literature on South African consumers perception and behaviour towards organic food. However, I have some concerns regarding the methodology, presentation of results and conclusions drawn.

Introduction

The introduction section should provide more meaningful insights into factors that affect intention to buy organic food (OF). There is an extensive body of literature on OF since it is a global phenomenon and much more would be expected in the introduction to the study even if literature on South African OF consumers is scarce.

Conceptual framework of the factors that influence the consumer purchase intentions for organically grown products

Conceptual framework is not well anchored in the relevant theories. Please provide adequate references.

Figure 2 is misleading and does not provide much insights into conceptual framework, please review it and relate it explicitly to your own research and the measurement instrument used in the study.

Materials and Methods

 The structure of the questionnaire including the questions and the scales used in the study should be precisely described to support interpretation of the model and overall results.

Did the Author verify if consumers understand the term “organic food” and how they interpret it?

Where there any other questions asked to the respondents?

How they identify/recognize organic food?

Study population, sample size and procedure

The sampling procedure is not convincingly demonstrated. I am not truly convinced that the Authors used systematic random sampling, i.e. selecting every 5th OGP consumer with equal chance of getting selected in the sample? Did all the customers declare to buy organic food? This section lacks some clarity and should be revised.

Moreover, the Authors should reflect on the potential consequences and bias resulting from small sample size even if the sample targets exclusively organic food consumers.

Results

The results are to some extent contradictory to the findings in literature on organic food consumers’ behaviour and attitudes.

The Authors claim that “the variable perception that OGPs have a better taste and of high quality was found to be statistically significant at 1% level (p = 0.000) and negatively correlated with the purchase intentions of OGPs”. Perception of organic food quality strongly affects intention to buy organic food that is confirmed in the literature. Consumers are unanimously positive about organic food quality that is a major driver of their purchasing decisions. Another issue is the employment status. The Authors admit that throughout literature employment status is reported to be positively correlated with the consumer purchase intentions of OGPs that is not confirmed in their study. Some reflection on it would be essential.

The small sample size bias could affect the reliability of the model and the overall results. I would suggest the Authors to review the results once again and reconsider if any other approach to data analysis could enhance reliability of the results and provide more meaningful insights.

Conclusions

The Authors claim that “The findings can be used for OGPs marketing strategies by accredited retail outlets, as well as inform consumers about the benefits of consuming OGPs” but this is not well justified.

The managerial implications should be clearly stated. The Authors should also relate their results to public health domain.

Majority of respondents were in the two lowest income categories that is not further elaborated. The Authors should reflect on it since the income level is a variable that affect intention to buy OF with some cross cultural differences reported in the literature not to be ignored.

I would suggest the Authors to refer to e.g. the review of Rödiger and Hamm (2015) to interpret their results regarding organic food prices and review the conclusions drawn.

Other comments:

to quote e.g.  “Source: Authors (2018)” or “Source: Survey data (2017/18)” or “Computed from STATA version 14” seem rather odd.

Author Response

.

Round 2

Reviewer 2 Report

The paper is significantly improved but the abstract need to be revised to provide more concise overview of the study and summary of the main findings. 

Author Response

.
